# Absence of *mgrB* Alleviates Negative Growth Effects of Colistin Resistance in *Enterobacter cloacae*

**DOI:** 10.3390/antibiotics9110825

**Published:** 2020-11-19

**Authors:** Jessie E. Wozniak, Aroon T. Chande, Eileen M. Burd, Victor I. Band, Sarah W. Satola, Monica M. Farley, Jesse T. Jacob, I. King Jordan, David S. Weiss

**Affiliations:** 1Emory Vaccine Center, Atlanta, GA 30317, USA; jewozni@emory.edu (J.E.W.); victor.band@alumni.emory.edu (V.I.B.); 2School of Medicine, Emory University, Atlanta, GA 30322, USA; eburd@emory.edu (E.M.B.); ssatola@emory.edu (S.W.S.); mfarley@emory.edu (M.M.F.); jtjacob@emory.edu (J.T.J.); 3Emory Antibiotic Resistance Center, Atlanta, GA 30329, USA; 4Institute for Bioengineering and Biosciences, Georgia Institute of Technology, Atlanta, GA 30332, USA; arch@gatech.edu; 5School of Biological Sciences, Georgia Institute of Technology, Atlanta, GA 30332, USA; king.jordan@biology.gatech.edu; 6Applied Bioinformatics Laboratory, Atlanta, GA 30346, USA; 7PanAmerican Bioinformatics Institute, Cali 760043, Valle del Cauca, Colombia; 8Atlanta VA Medical Center, Decatur, GA 30033, USA; 9IHRC Applied Bioinformatics Laboratory, Georgia Institute of Technology, Atlanta, GA 30332, USA

**Keywords:** colistin, *Enterobacter*, CRE

## Abstract

Colistin is an important last-line antibiotic to treat highly resistant *Enterobacter* infections. Resistance to colistin has emerged among clinical isolates but has been associated with a significant growth defect. Here, we describe a clinical *Enterobacter* isolate with a deletion of *mgrB*, a regulator of colistin resistance, leading to high-level resistance in the absence of a growth defect. The identification of a path to resistance unrestrained by growth defects suggests colistin resistance could become more common in *Enterobacter*.

## 1. Introduction

Antibiotic resistance is an impending crisis that threatens advancements of modern medicine including transplants, cancer therapy, and common surgical procedures such as joint replacements. Recent reports estimate that 2.8 million people are diagnosed with an antibiotic-resistant infection each year in the United States, leading to between 35,000 and 160,000 deaths [1,2]. Carbapenem-resistant *Enterobacterales* (CRE) including *Enterobacter*, *Klebsiella*, and *Escherichia spp*. are associated with a mortality rate of up to 50% [3]. These bacterial infections are only susceptible to a handful of “last line” drugs such as colistin. Unfortunately, resistance to colistin is increasingly observed.

A common mechanism of colistin resistance in *Enterobacterales* involves chromosomal mutations in the two-component system (TCS) PhoPQ or in genes in its regulon. PhoQ is a sensor kinase that responds to polymyxin antibiotics including colistin, as well as cationic antimicrobial peptides and environmental magnesium and calcium [4,5]. When activated, PhoQ phosphorylates its response regulator, PhoP. Phosphorylated PhoP is a transcriptional regulator with a broad regulon of hundreds of genes, which includes its self-regulation of the *phoPQ* operon [6]. An additional level of regulation of the PhoPQ system is mediated by the inner membrane protein and negative regulator, MgrB [4,7]. The deletion of *mgrB* results in an upregulation of *phoPQ* transcription and the overexpression of *mgrB* decreases *phoPQ* transcription, regardless of magnesium concentration [7]. The mechanism(s) by which MgrB regulates PhoPQ is not clear, but the protein may directly interact with PhoQ in the inner membrane [7].

PhoPQ regulates biosynthetic operons involved in the modification of the lipid A portion of lipopolysaccharide (LPS), the major structural component of the Gram-negative outer membrane. PhoPQ activation leads to the addition of cationic sugars to lipid A, which increase the charge of the outer membrane. In *Enterobacter cloacae,* PhoPQ is required for the addition of positively charged 4-amino-4-deoxy-L-arabinose (L-Ara4N) to lipid A, mediated by *arnBCADTEF* [8,9]. This increased positive charge leads to the electrostatic repulsion of colistin, which is also cationic, and results in bacterial resistance.

Lipid A modifications occur in *Enterobacterales* and numerous other bacteria. In *Pseudomonas aeruginosa,* activating point mutations in PhoPQ and the PmrAB TCS are implicated in colistin resistance [10]. Although *pmrAB* are encoded in *Enterobacter cloacae,* these genes were not found to be involved in colistin resistance [9]. The roles of PhoPQ and PmrAB in lipid A modification and resistance to cationic antimicrobial peptides have been extensively reviewed elsewhere [6,10].

Previous studies indicated that colistin resistance is often associated with a reduced growth rate (a biological fitness cost) [11,12] and that in the absence of colistin, these mutations can revert and lead to the loss of resistance [13,14]. Fitness costs associated with colistin resistance (by multiple mechanisms) have been well documented in the *Enterobacterales* species *Escherichia coli* and *Klebsiella pneumoniae*, as well as in *P. aeruginosa* and *Acinetobacter baumannii* [15,16,17]. In particular, mutations in PhoPQ that lead to constitutive lipid A modification and colistin resistance exert a fitness cost and are reversible in the absence of colistin [13]. Additionally, mutations in *pmrAB* are associated with costly and reversible colistin resistance in *A. baumannii* [14].

In contrast, the inactivation of *mgrB* is a known mechanism of fitness cost-free colistin resistance in *K. pneumoniae,* although the reason it does not exert a fitness cost is unknown [16]. Mutations within *mgrB* leading to colistin resistance have very recently been described in *Enterobacter* species, but the effect these mutations have on the growth rate has not been interrogated [18,19].

This study details a colistin-resistant *Enterobacter cloacae* clinical isolate lacking *mgrB* and reveals that this path to resistance occurs without the characteristic negative effect on growth.

## 2. Results

We identified a colistin-resistant *Enterobacter cloacae* clinical isolate, Mu471, collected from the Georgia Emerging Infections Program’s Multi-site Gram-negative Surveillance Initiative (Figure 1A) [20]. Bioinformatic analysis revealed an approximately 4 kb region absent from the Mu471 chromosome in comparison to strain ATCC13047 [21], a close genetic relative to Mu471 in the NCBI database. Average nucleotide identity (ANI) between ATCC13047 and Mu471 and between Mu471 and *Enterobacter cloacae* DSM 30054 was >98%, confirming Mu471 to be *E. cloacae.* Interestingly, among the genes encoded in this 4 kb region was *mgrB*, a negative regulator of PhoPQ (Appendix A). To determine if the absence of *mgrB* was responsible for colistin resistance in Mu471, we introduced a plasmid (pBAV1K-T5-gfp) encoding *mgrB* from ATCC13047 into this strain, creating strain Mu471 p*mgrB* [21,22]. Colistin susceptibility testing was performed by Etest (bioMérieux, Marcy-l’Étoile, France) (Figure 1A) and broth microdilution (BMD; Appendix A) [8]. Indeed, the expression of *mgrB* in Mu471 p*mgrB* led to colistin susceptibility (Figure 1B), indicating that the loss of *mgrB* was responsible for the colistin resistance of the parental strain (Figure 1A). Importantly, there was no difference in the growth curve of Mu471 pBAV (empty vector) and Mu471 p*mgrB*, highlighting that the colistin resistance of Mu471 did not have a negative effect on the growth rate (Figure 1C). To our knowledge, this is the first demonstration of colistin resistance in *Enterobacter cloacae* in the absence of an observable effect on growth.

Since the complementation of *mgrB* into a colistin-resistant strain that lacks this gene resulted in susceptibility, we then tested whether the deletion of this gene would lead to broad resistance in a strain with only a minor population of resistant cells (ATCC13047). The sensitivity of the Etest revealed that ATCC13047 is largely composed of susceptible cells, resulting in a large zone of clearing, however, a minor subpopulation of resistant cells was observed as well (Figure 2A). The deletion of *mgrB* from ATCC13047 (creating strain ATCC13047 Δ*mgrB*) led to a vast expansion of the resistant cells, and the strain exhibited no zone of clearing by Etest (Figure 2A,B). Due to the presence of resistant cells in both cases, BMD was not sensitive enough to detect the differences in susceptibility between ATCC13047 and ATCC13047 Δ*mgrB* (Appendix A). Most importantly, in agreement with the results from Mu471 (Figure 1A,B), the deletion of *mgrB* and the subsequent increase in colistin resistant cells in ATCC13047 did not correspond to a decrease in growth rate (Figure 2C). Taken together, these results suggest that the absence of *mgrB* is a previously unobserved path to high-level colistin resistance without an observable impact on growth in *E. cloacae*.

## 3. Discussion

To our knowledge, this is the first identification of an *Enterobacter cloacae* clinical isolate (Mu471) that naturally lacks *mgrB* and subsequently displays high-level colistin resistance without an observable impact on growth rate. Introduction of *mgrB* into this strain reversed colistin resistance (Figure 1B). Additionally, when *mgrB* was deleted from a distinct strain, ATCC13047, this led to colistin resistance (Figure 2B).

Supporting these findings are recent reports of colistin resistance via *mgrB* inactivation in *Klebsiella pneumoniae*, which were also without an impact on growth [16,23,24,25]. Reports of this mechanism of resistance in *K. pneumoniae* are increasing, including a comprehensive study from Greece which found *mgrB* inactivation to be the most common mechanism of colistin resistance and deemed this phenomenon an emerging epidemic [15,26].

## 4. Materials and Methods

### 4.1. Bacterial Growth Conditions

Strains used in this study included Mu471, ATCC13047, and ATCC13047 Δ*mgrB* (Table 1). Plasmids used in this study were pBAV (derived from pBAVIK-t5-GFP [22]) empty vector and pBAV *mgrB* (Table 2). pBAV *mgrB* was generated using the primers below with the *mgrB* coding region from ATCC13047 inserted to pBAVIK-t5-GFP between the EcoRI and PstI sites (Table 3). All liquid cultures were grown in Mueller–Hinton broth at 37 °C overnight unless otherwise stated. Etests were performed in accordance with bioMérieux on Mueller–Hinton agar plates with kanamycin (90 μg/mL) when plasmid retention was necessary. Broth microdilution was performed in accordance with the CLSI-EUCAST (Clinical and Laboratory Standards Institute and European Committee on Antimicrobial Susceptibility Testing) joint polymyxin breakpoints working group [27].

### 4.2. Cloning

The construction of plasmid for complementation was performed as previously described in a pBAVIK-T5-GFP derivative [8]. Deletion of *mgrB* in ATCC13047 was also performed as previously described. Briefly, *mgrB* was replaced with a kanamycin resistance cassette (KanFRT) by λ Red mutagenesis [28], resulting in ATCC13047 *mgrB*::KanFRT. A clean deletion strain was subsequently generated by the introduction of the vector pCP20 containing the FLP recombinase proteins, resulting in strain ATCC13047 Δ*mgrB* [29]. Mutations were confirmed by PCR.

### 4.3. Growth Curves

Bacterial growth was measured by optical density (OD_600_) using a plate reader (Biotek Synergy MX plate reader (Applied Biosystems)). Bacteria were grown at 37 °C shaking at 220 rpm for 24 h as previously described [30]. Bacteria were grown in Mueller–Hinton broth with kanamycin (90 μg/mL) for plasmid retention when necessary.

### 4.4. Bioinformatics

The integrated genome visualization of a read to genome mapping using ATCC13047 as a reference sequence (NCBI Reference Sequence: NC_014121.1) was performed. (Mu471 sequence runs: SRR4035130). Bwa version 0.7.17 (r1188), https://github.com/lh3/bwa, was used for mapping. Visualization was done with samplot version 1.0.14, https://github.com/jbelyeu/samplot, and edited in Illustrator. Average Nucleotide Identity (ANI) was calculated and Mu471 has an ANI value of >98% compared to ATCC13047, and >98% compared to *E. cloacae* strain DSM 30054. >9 ANI was performed using mummer v3 [31].

## 5. Conclusions

Multiple studies demonstrate that the cost of resistance to colistin can be so great that the reversal of this phenotype occurs in the absence of the antibiotic. In contrast, the evolution of colistin resistance without an impact on growth may lead to more rapid increases in the frequency of colistin resistant isolates, especially if colistin usage increases in the clinic.

## Figures and Tables

**Figure 1 antibiotics-09-00825-f001:**
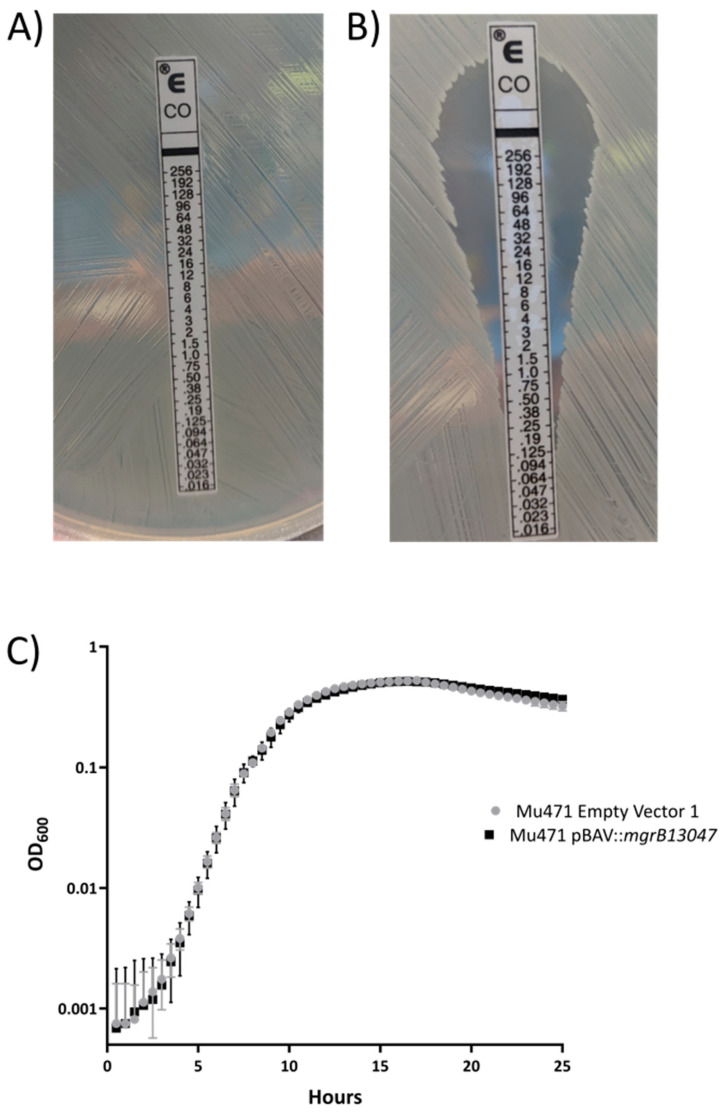
Mu471 exhibits high-level colistin resistance in the absence of a growth delay. (**A**) Etest of Mu471. (**B**) Etest of Mu471 with pBAV P*mgrB*. Agar contains kanamycin (90 μg/mL) for plasmid retention. (**C**) Growth curves of Mu471 with pBAV P*mgrB* and empty vector control. Growth curves were performed in Mueller–Hinton broth with kanamycin (90 μg/mL) for plasmid retention.

**Figure 2 antibiotics-09-00825-f002:**
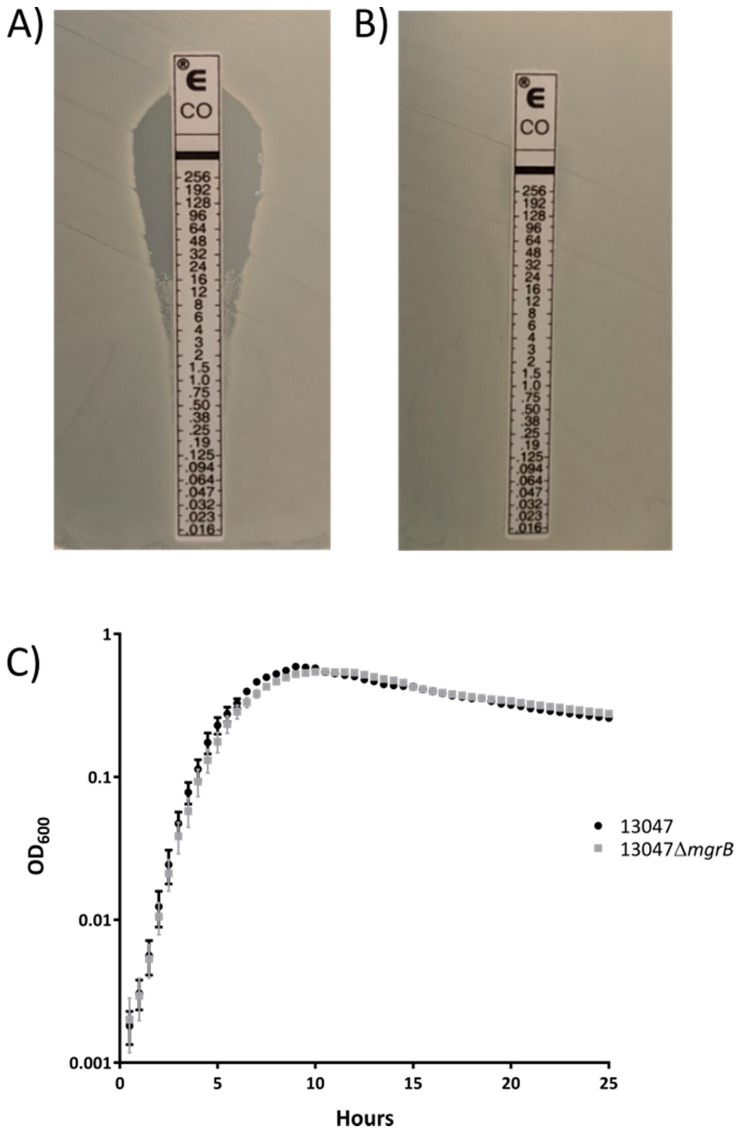
Deletion of *mgrB* causes high-level colistin resistance in the absence of a growth delay. (**A**) Etest of ATCC13047. (**B**) Etest of ATCC13047Δ*mgrB*. (**C**) Growth curve of ATCC13047 and ATCC13047 Δ*mgrB*. Growth curves were performed in Mueller–Hinton broth.

**Table 1 antibiotics-09-00825-t001:** Bacterial strains used in this study.

Strain Name	Species	Strain Information
Mu471	*Enterobacter cloacae*	Collected by MuGSI from the urine of a female patient from a GA hospital in 2013
ATCC13047	*Enterobacter cloacae*	Purchased from ATCC
ATCC13047Δ*mgrB*	*Enterobacter cloacae*	Clean *mgrB* deletion generated as described below

**Table 2 antibiotics-09-00825-t002:** Plasmids used in this study.

Plasmid Name	Information
pBAV EV	Derived from pBAVIK-t5-GFP
pBAV *mgrB*	Complementation plasmid containing *mgrB* generated as described below

**Table 3 antibiotics-09-00825-t003:** Primers used in this study.

Primer Name	Sequence
pBAV *mgrB* F	ccacaattatgatagaatttgacgtcgaattccattgcctctttatctttgttgtcatgc
pBAV *mgrB* R	gcggcggcatcgatcgggccctgaggcctgcagttcaccacctcaataaaaacacgc

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
