# Peer review of "Absence of mgrB Alleviates Negative Growth Effects of Colistin Resistance in Enterobacter cloacae"

_antibiotics, 2020, doi:10.3390/antibiotics9110825_

Round 1

Reviewer 1 Report

In this manuscript, the authors describe an isolate of Enterobacter cloacae that was highly resistant to colistin due to loss of mgrB gene.

Comments:

1.  This is not the first description of mgrB inactivation in Enterobacter, please see PMIDs: 32253218 and 31838239. Please remove all references to first description.  

2.  It is not clear which reference shows that the absence of mgrB leads to a growth deficit in Enterobacter?  Please provide this.  The authors reference work from A. baumannii, however the genes involved are not mgrB.  They also reference K. pneumoniae, but that was mcr-1, and also P. aeruginosa for the reversion phenotype.  Where are the data for Enterobacter?    They need to tie the story better together in the introduction.  Please add more details. Under what circumstances does colistin resistance lead to a growth deficit and in which bacteria? Under what circumstances does colistin resistance NOT lead to a growth deficit and in what bacteria?  The deletion of mgrB in K. pneumoniae doesn't affect growth and Kpn and Enterobacter are at least both Enterobacteriaceae.  The impact of this study is not clear. 

3.  Line 42 refers to "PhoPQ mutations" Do the authors mean alterations in the PhoPQ regulon? 

Author Response

We thank the reviewer for their thoughtful comments which we feel have allowed us to significantly improve the quality of the manuscript.

Comments:

  1. This is not the first description of mgrB inactivation in Enterobacter, please see PMIDs: 32253218 and 31838239. Please remove all references to first description.  

We thank the reviewer for highlighting these publications and apologize for not including them previously. We have now included these references in the introduction and altered the wording throughout to accurately reflect the novel aspects of our findings. These additions have been made at lines 67-69:

“Mutations within mgrB leading to colistin resistance have very recently been described in Enterobacter species, but the effect these mutations have on growth rate has not been interrogated [18, 19].”

  1. It is not clear which reference shows that the absence of mgrB leads to a growth deficit in Enterobacter?  Please provide this.  The authors reference work from A. baumannii, however the genes involved are not mgrB.  They also reference K. pneumoniae, but that was mcr-1, and also P. aeruginosa for the reversion phenotype.  Where are the data for Enterobacter?    They need to tie the story better together in the introduction.  Please add more details. Under what circumstances does colistin resistance lead to a growth deficit and in which bacteria? Under what circumstances does colistin resistance NOT lead to a growth deficit and in what bacteria?  The deletion of mgrB in K. pneumoniae doesn't affect growth and Kpn and Enterobacter are at least both Enterobacteriaceae.  The impact of this study is not clear. 

We apologize for our lack of clarity, and have clarified the introduction to make several points. First, colistin resistance is known to be associated with a fitness cost in Pseudomonas aeruginosa, Escherichia coli, and Acinetobacter baumannii. In P. aeruginosa and E. coli, colistin resistance is dependent on the PhoPQ two-component system. In A. baumannii, colistin resistance is dependent on the PmrAB two-component system and independent of mgrB since this gene is not encoded in the genome. This highlights that the fitness cost of colistin resistance in A. baumannii cannot be dependent on mgrB. Second, absence of mgrB is not known to result in a growth deficit; rather absence or inactivation of mgrB in Klebsiella pneumoniae has been shown to result in colistin resistance without a fitness cost. The impact of the current study is in 1) describing the first instance of mgrB deletion in a naturally occurring clinical isolate of Enterobacter, and 2) demonstrating for the first time that the absence of mgrB in Enterobacter causes colistin resistance without a fitness cost. Neither of the papers highlighted by the reviewer in comment 1 investigated the effect of mgrB on fitness cost. We feel this is a critical point. Since fitness costs are thought to limit the emergence and stability of antibiotic resistance in bacterial populations, the fact that we have shown that Enterobacter has a pathway to colistin resistance without a fitness cost suggests that colistin resistance may increase in this pathogen. Furthermore, these findings suggest that clinical microbiology laboratories and surveillance networks should be on the lookout for increased emergence of colistin resistance in Enterobacter. More clarification has been added to the introduction with the hopes of clarifying the impact of the study, namely lines 53-57 and 60-71:

“53-57: “Lipid A modifications occur in Enterobacterales and numerous other bacteria. In Pseudomonas aeruginosa, activating point mutations in PhoPQ and the PmrAB TCS are implicated in colistin resistance [10]. Although pmrAB are encoded in Enterobacter cloacae, these genes were not found to be involved in colistin resistance [9]. The roles of PhoPQ and PmrAB in lipid A modification and resistance to cationic antimicrobial peptides have been extensively reviewed elsewhere [6, 10]”

60-71: “Fitness costs associated with colistin resistance (by multiple mechanisms) have been well documented in the Enterobacterales species Escherichia coli and Klebsiella pneumoniae, as well as in Pseudomonas aeruginosa and Acinetobacter baumannii [15-17]. In particular, mutations in PhoPQ that lead to constitutive lipid A modification and colistin resistance exert a fitness cost and are reversible in the absence of colistin [13]. Additionally, mutations in pmrAB are associated with costly and reversible colistin resistance in A. baumannii [14].

In contrast, inactivation of mgrB is a known mechanism of fitness cost-free colistin resistance in K. pneumoniae, although the reason it does not exert a fitness cost is unknown [16]. Mutations within mgrB leading to colistin resistance have very recently been described in Enterobacter species, but the effect these mutations have on growth rate has not been interrogated [18, 19].

This study details a colistin-resistant Enterobacter cloacae clinical isolate lacking mgrB and reveals that this path to resistance occurs without the characteristic negative effect on growth.”

  1. Line 42 refers to "PhoPQ mutations" Do the authors mean alterations in the PhoPQ regulon? 

The mention of PhoPQ mutations was meant to imply point mutations in the PhoPQ coding sequence, but mutations in the regulon (i.e. arnBCEDTEF) can also be responsible for colistin resistance. We have expanded on the regulon in lines 36-40:

“PhoQ is a sensor kinase that responds to polymyxin antibiotics including colistin, as well as cationic antimicrobial peptides and environmental magnesium and calcium [4, 5]. When activated, PhoQ phosphorylates its response regulator, PhoP. Phosphorylated PhoP is a transcriptional regulator with a broad regulon of hundreds of genes, which includes its self-regulation of the phoPQ operon [6].”

And lines 49-52:

“In Enterobacter cloacae, PhoPQ is required for the addition of positively charged 4-amino-4-deoxy-L-arabinose (L-Ara4N) to lipid A, mediated by arnBCADTEF [8, 9]. This increased positive charge leads to electrostatic repulsion of colistin, which is also cationic, and results in bacterial resistance.”

Reviewer 2 Report

The authors describe a study aimed to show evidences that a natural delection of mgrB in a clinical Enterobacter spp. isolates leads to high-level of colistin resistance, without an observable impact on growth. Moreover, the study also point that insertion of mgrB in this strain reverse the phenotype of colistin resistance. Although this is an important description, the study has some important weaknesses that should be clarified including:

1. Introduction.

1.1. There are two-component system-mediated LPS modifications but only PhoP/PhoQ was referred by the authors (Line 37). 

1.2. What were the bacterial species that present PhoPQ mutations associated with a reduced growth, or PhoPQ mutations that can be reverted and lead to the loss of resistance in the absence of colistin? 

2. Results

2.1 How was the isolate identified at species level? Did you take into consideration at least ANI (Average Nucleotide Identity) between strain Mu471 and type strains of Enterobacter genus? Information regarding year of isolation and origin should be included.

2.2. Did the authors checked that there were no other mechanisms of colistin resistance (e.g. prmA or prmB mutations) besides the lack of mgrB that could lead to colistin resistance in strain Mu471? 

2.3. How the authors confirmed that the mgrB was in fact delected in the strain 13047 ΔmgrB (Line 66)? Did you performed pcr or sequenced the strain?

3. Discussion

3.1. The authors stated that the strain Mu471 is carbapenem resistant but there is no proofs of that (Line 85).

4. Materials and Methods

4.1. Accurate bacterial identification should be included in this section (Line 97).

4.2. The colistin MIC determination must be performed by the ISO-standard broth microdilution method (20776-1), which was agreed by the CLSI-EUCAST joint Polymyxin Breakpoints Working Group (https://www.eucast.org/fileadmin/src/media/PDFs/EUCAST_files/General_documents/Recommendations_for_MIC_determination_of_colistin_March_2016.pdf) (Lines 100-101).

4.3. What tools or software were used to perform genome mapping? (Lines 113-116).

4.4. Sequence run SRR4035130 reffers to isolate MUGSI_228 and not to Mu471 (Lines 115-116)-

Minor comments:

  • Line 50: Here and through the text replace Enterobacter cloacae by the abbreviation E. cloacae and place all the bacterial species name in italic.
  • Line 54: Here and through the text "mgrB" gene should be in italic.
  • Line 67: Replace 13047 by Strain 13047.

Author Response

We thank the reviewer for numerous critically important points which we feel have allowed us to significantly improve the quality of the manuscript.

The authors describe a study aimed to show evidences that a natural delection of mgrB in a clinical Enterobacter spp. isolates leads to high-level of colistin resistance, without an observable impact on growth. Moreover, the study also point that insertion of mgrB in this strain reverse the phenotype of colistin resistance. Although this is an important description, the study has some important weaknesses that should be clarified including:

  1. Introduction.

1.1. There are two-component system-mediated LPS modifications but only PhoP/PhoQ was referred by the authors (Line 37). 

This is an important point highlighted by the reviewer. There are indeed several two-component systems that can play a role in LPS modifications. We believe that PhoPQ is the most critical, even though Enterobacter cloacae also encodes PmrAB. This is in part because a pmrAB mutant was generated in E. cloacae in a study by Kang KN et al (PMID: 30873646) and was shown to have no effect on colistin resistance. In contrast, lipid A modifications and colistin resistance are abolished in a phoPQ mutant (Band et al, PMID: 27572838) suggesting that PhoPQ is a critical TCS involved in lipid A modification in (arnBCADTEF) E. cloacae. We have now expanded the Introduction to include a description of PmrAB and we also direct the reader to two thorough review articles on the subject of LPS modification as indicated below (lines 53-57):

“Lipid A modifications occur in Enterobacterales and numerous other bacteria. In Pseudomonas aeruginosa, activating point mutations in PhoPQ and the PmrAB TCS are implicated in colistin resistance [10]. Although pmrAB are encoded in Enterobacter cloacae, these genes were not found to be involved in colistin resistance [9]. The roles of PhoPQ and PmrAB in lipid A modification and resistance to cationic antimicrobial peptides have been extensively reviewed elsewhere  [6, 10].”

1.2. What were the bacterial species that present PhoPQ mutations associated with a reduced growth, or PhoPQ mutations that can be reverted and lead to the loss of resistance in the absence of colistin? 

Resistance due to mutations in PhoPQ associated with fitness costs has been observed in Lee J et al (pmid: 27150578), wherein the authors evolved multiple lineages of a strain of P. aeruginosa to become resistant to colistin. One of the resistant lineages had an amino acid change in PhoQ. When antibiotic selection pressure was removed, the colistin resistance phenotype was reversed. Additionally, several strains with phoPQ chromosomal mutations were shown to be less competitively fit in E. coli (pmid: 32477288). Fitness costs associated with colistin resistance in Acinetobacter baumannii are due to mutations in PmrAB, and A. baumannii does not encode PhoPQ (pmid: 21216865, 24247145). These studies are now referenced in the expanded Introduction section:

“Previous studies indicated that colistin resistance is often associated with a reduced growth rate (a biological fitness cost) [11, 12] and that in the absence of colistin, these mutations can revert and lead to the loss of resistance [13, 14]. Fitness costs associated with colistin resistance (by multiple mechanisms) have been well documented in the Enterobacterales species Escherichia coli and Klebsiella pneumoniae, as well as in Pseudomonas aeruginosa and Acinetobacter baumannii [15-17]. In particular, mutations in PhoPQ that lead to constitutive lipid A modification and colistin resistance exert a fitness cost and are reversible in the absence of colistin [13]. Additionally, mutations in pmrAB are associated with costly and reversible colistin resistance in A. baumannii [14].” (lines 58-65)

  1. Results

2.1 How was the isolate identified at species level? Did you take into consideration at least ANI (Average Nucleotide Identity) between strain Mu471 and type strains of Enterobacter genus? Information regarding year of isolation and origin should be included.

Mu471 was confirmed to be Enterobacter by MALDI-TOF analysis in the Emory University Hospital Clinical Microbiology Laboratory. In addition, at the request of the reviewer, ANI was performed and Mu471 has an ANI value of >99% compared to Enterobacter cloacae subsp. cloacae strain CAPREX_E7, which is the Enterobacter cloacae representative isolate / type strain at NCBI. ANI was performed using mummer v3. This has been included at lines 155-157. Furthermore, additional information about the strain is now included in the paper: “Collected by MuGSI from the urine of a female patient from a GA hospital in 2013.” (Line 128).

2.2. Did the authors checked that there were no other mechanisms of colistin resistance (e.g. prmA or prmB mutations) besides the lack of mgrB that could lead to colistin resistance in strain Mu471? 

Our data demonstrate that addition of mgrB alone is sufficient to reverse colistin resistance in Mu471. Since MgrB acts on the PhoPQ pathway, it is likely that if a PhoPQ-independent pathway (like PmrAB) also played a significant role in colistin resistance in this strain, addition of mgrB alone would not have reversed the resistance phenotype. Additionally, as mentioned above, Kang et al (PMID: 30873646) showed that PmrAB is not involved in colistin resistance in Enterobacter, while Band et al (PMID: 27572838) showed that resistance is dependent on PhoPQ. Taken together, these data indicate that PhoPQ is the primary colistin resistance pathway in Enterobacter.

An additional line was included in the manuscript “Although pmrAB are encoded in Enterobacter cloacae, these genes were not found to be involved in colistin resistance” (line 55-56).

2.3. How the authors confirmed that the mgrB was in fact delected in the strain 13047 ΔmgrB (Line 66)? Did you performed pcr or sequenced the strain?

The absence of mgrB in strain 13047ΔmgrB was confirmed by PCR. The presence of the kanamycin resistance cassette used for cloning was confirmed to be present in 13047 mgrB::KanR in the location in the genome previously occupied by mgrB. Subsequently, in strain 13047ΔmgrB, both mgrB and the kanamycin resistance cassette were absent and undetectable by PCR.

More clarification has been added to the methods section: “Mutations were confirmed by PCR.” (line 145)

  1. Discussion

3.1. The authors stated that the strain Mu471 is carbapenem resistant but there is no proofs of that (Line 85).

We are very grateful to the reviewer for this comment which allowed us to catch this serious mistake. Strain Mu471 was collected as a part of the Multi-site Gram-Negative Surveillance initiative, which seeks to monitor carbapenem-resistant bacteria. After performing BMD for carbapenem antibiotics at the request of the reviewer, Mu471 was in fact found to be susceptible to the carbapenem antibiotics tested (meropenem, imipenem, doripenem). We have now updated the manuscript and title to reflect these new data.

  1. Materials and Methods

4.1. Accurate bacterial identification should be included in this section (Line 97).

A more expansive bacterial identification has been included at the request of the reviewer:

“Collected by MuGSI from the urine of a female patient from a GA hospital in 2013.” (line 128)

“Average Nucleotide Identity (ANI) was calculated and Mu471 has an ANI value of >99% compared to Enterobacter cloacae subsp. cloacae strain CAPREX_E7, which is the Enterobacter cloacae representative isolate / type strain at NCBI. ANI was performed using mummer v3.” (Lines 155-157)

4.2. The colistin MIC determination must be performed by the ISO-standard broth microdilution method (20776-1), which was agreed by the CLSI-EUCAST joint Polymyxin Breakpoints Working Group (https://www.eucast.org/fileadmin/src/media/PDFs/EUCAST_files/General_documents/Recommendations_for_MIC_determination_of_colistin_March_2016.pdf) (Lines 100-101).

Broth microdilution (BMD) was performed at the request of the reviewer: the MICs are as follows, Mu471 > 200μg/mL, Mu471 pMgrB = 0.78 μg/mL, Strain 13047 > 200μg/mL, Strain 13047ΔmgrB > 200μg/mL. The results of the BMD are now included in Supplemental Table 1 and mentioned at several points within the manuscript.

4.3. What tools or software were used to perform genome mapping? (Lines 113-116).

Bwa version 0.7.17 (r1188), https://github.com/lh3/bwa was used for mapping.  Visualization was done with samplot version 1.0.14, https://github.com/jbelyeu/samplot, and cleaned up in Illustrator.

This information has now been included in the methods section of the paper for clarity (lines 153-155).

4.4. Sequence run SRR4035130 reffers to isolate MUGSI_228 and not to Mu471 (Lines 115-116)-

We apologize for the confusion and that the isolate numbers are not more clear. Mu471 was acquired from the Georgia Emerging Infections Program (EIP) before it was sequenced by MuGSI at the CDC. Not all of the Georgia EIP isolates are eventually chosen to be included in the MuGSI program and sequenced. Therefore, the Georgia EIP isolate numbers (i.e. Mu471) often do not match the MuGSI CDC isolate numbers (i.e. MUGSI_228). Please be assured that Mu471 and MUGSI_228 are in fact the same.

Minor comments:

  • Line 50: Here and through the text replace Enterobacter cloacaeby the abbreviation  cloacaeand place all the bacterial species name in italic.
  • Line 54: Here and through the text "mgrB" gene should be in italic.
  • Line 67: Replace 13047 by Strain 13047.

These changes have been made, thank you.

Reviewer 3 Report

The brief report article of Wozniak et al describes an interesting finding regarding the discovery that the inactivation of mgrB gene is responsible of colistin resistance in Enterobacter cloacae.

The manuscript is well written and the scope of the authors is clear. The proof of the involvement of mgrB gene in colistin resistance is well documented by different experiments through the introduction of a pmrgB in a resistant strain that became susceptible to colistin and also by deleting mgrB gene in a susceptible strain that became resistant.

However, novelty can be considered only regarding Enterobacter. Indeed, this finding was already known regarding other Gram-negative pathogens. Thus, I suggest to the authors to introduce references about this in the introduction. I can suggest:

  • Inactivation of mgrB gene regulator and resistance to colistin is becoming endemic in carbapenem-resistant Klebsiella pneumoniae in Greece: A nationwide study from 2014 to 2017
  • MgrB Inactivation Is a Common Mechanism of Colistin Resistance in KPC-Producing Klebsiella pneumoniae of Clinical Origin
  • Unravelling of a mechanism of resistance to colistin in Klebsiella pneumoniae using atomic force microscopy
  • Preservation of Acquired Colistin Resistance in Gram-Negative Bacteria

Author Response

The brief report article of Wozniak et al describes an interesting finding regarding the discovery that the inactivation of mgrB gene is responsible of colistin resistance in Enterobacter cloacae.

The manuscript is well written and the scope of the authors is clear. The proof of the involvement of mgrB gene in colistin resistance is well documented by different experiments through the introduction of a pmrgB in a resistant strain that became susceptible to colistin and also by deleting mgrB gene in a susceptible strain that became resistant.

However, novelty can be considered only regarding Enterobacter. Indeed, this finding was already known regarding other Gram-negative pathogens. Thus, I suggest to the authors to introduce references about this in the introduction. I can suggest:

  • Inactivation of mgrB gene regulator and resistance to colistin is becoming endemic in carbapenem-resistant Klebsiella pneumoniae in Greece: A nationwide study from 2014 to 2017
  • MgrB Inactivation Is a Common Mechanism of Colistin Resistance in KPC-Producing Klebsiella pneumoniae of Clinical Origin
  • Unravelling of a mechanism of resistance to colistin in Klebsiella pneumoniae using atomic force microscopy
  • Preservation of Acquired Colistin Resistance in Gram-Negative Bacteria

Response:

Thank you for the suggestions of additional references to add clarity to the manuscript. These references have been included in the manuscript to add context to the Introduction. We hope that these revisions will clarify the novelty and impact of this study.

The references have been added at lines 58-60:

“Previous studies indicated that colistin resistance is often associated with a reduced growth rate (a biological fitness cost) [11, 12] and that in the absence of colistin, these mutations can revert and lead to the loss of resistance [13, 14].”

And 64-65:

“Additionally, mutations in pmrAB are associated with costly and reversible colistin resistance in A. baumannii [14].”

Finally, lines 66-69:

“In contrast, inactivation of mgrB is a known mechanism of fitness cost-free colistin resistance in K. pneumoniae, although the reason it does not exert a fitness cost is unknown [16]. Mutations within mgrB leading to colistin resistance have very recently been described in Enterobacter species, but the effect these mutations have on growth rate has not been interrogated [18, 19].”

Round 2

Reviewer 1 Report

The authors have sufficiently addressed my previous comments.  

Author Response

The authors thank the reviewer for their critical reading of this manuscript 

Reviewer 2 Report

The authors substantially improve the article. However, there are still some critical issues related to the methodology and results that must be analyzed/improved.

Bacterial Identification:  - Lines 155-156: "Mu471 has an ANI value of 155 >99% compared to Enterobacter cloacae subsp. cloacae strain CAPREX_E7". This is a result and should be included in results section.

- ANI determination should include a type strain of E. cloacae subsp. cloacae, which is not the strain CAPREX_E7. In this sense, the ANI values between strain Mu471 and an authentic type strain of E. cloacae subsp. cloacae should be determined. Type strains of Enterobacter cloacae subsp. cloacae include: NCDC 279-56, DSM 30054 (genome available at NCBI: NZ_CP056776.1), ATCC 13047, CIP 60.85, IFO 13535, NBRC 13535, NCTC 10005, and WDCM 00083. 

- Line 157: Include a reference to support "mummer v3."

Colistin susceptibility test:

- Again, the colistin susceptibility of all strains should be tested by BMD, and according to reference [28], and not by E-test. In this sense, eliminate all data concerning E-test methodology, results, discussion and Figures. Table S1 should be included in the main article.
- Figure 1C. What was the medium used and the colistin concentration tested to perform growth curves analysis? Reference 31 is not suitable (Line 149). Please check the paper from Giordano et al. mSphere 2019 (https://www.ncbi.nlm.nih.gov/pmc/articles/PMC6835208/) to be aware how these experiments should be performed or to check if the experiments were performed accordingly.

Author Response

The authors substantially improve the article. However, there are still some critical issues related to the methodology and results that must be analyzed/improved.

Bacterial Identification:  - Lines 155-156: "Mu471 has an ANI value of 155 >99% compared to Enterobacter cloacae subsp. cloacae strain CAPREX_E7". This is a result and should be included in results section.

- ANI determination should include a type strain of E. cloacae subsp. cloacae, which is not the strain CAPREX_E7. In this sense, the ANI values between strain Mu471 and an authentic type strain of E. cloacae subsp. cloacae should be determined. Type strains of Enterobacter cloacae subsp. cloacae include: NCDC 279-56, DSM 30054 (genome available at NCBI: NZ_CP056776.1), ATCC 13047, CIP 60.85, IFO 13535, NBRC 13535, NCTC 10005, and WDCM 00083. 

As suggested, we performed ANI between Mu471 and E. cloacae DSM 30054 and found >98% identity value. This has been reflected in the results section in lines 77-78:

“Average nucleotide identity (ANI) between ATCC13047 and Mu471 and between Mu471 and Enterobacter cloacae DSM 30054 was >98%, confirming Mu471 to be E. cloacae.”

We also clarified this in the materials and methods section at lines 159-161:

“Average Nucleotide Identity (ANI) was calculated and Mu471 has an ANI value of >98% compared to ATCC13047, and >98% compared to E. cloacae strain DSM 30054.”

- Line 157: Include a reference to support "mummer v3."

We have included a reference to support mummer v3 (line 162):

ANI was performed using mummer v3 [32].”

The reference is: " Kurtz, S., et al., Versatile and open software for comparing large genomes. Genome Biol, 2004. 5(2): p. R12”

Colistin susceptibility test:

- Again, the colistin susceptibility of all strains should be tested by BMD, and according to reference [28], and not by E-test. In this sense, eliminate all data concerning E-test methodology, results, discussion and Figures. Table S1 should be included in the main article.

We are grateful for the reviewer’s comments and previously added the requested BMD data. While we appreciate that BMD is a standard method for testing colistin susceptibility in a clinical setting, the goal of this manuscript is to determine the mechanistic explanation for the colistin resistance exhibited by Mu471, and then to solidify the role of mgrB deletion in colistin resistance that occurs without a negative effect on growth in Enterobacter. In this light, we feel that a visual representation of the phenotype of the different strains (by Etest) complements the BMD data and therefore favor including both methods.

- Figure 1C. What was the medium used and the colistin concentration tested to perform growth curves analysis? Reference 31 is not suitable (Line 149). Please check the paper from Giordano et al. mSphere 2019 (https://www.ncbi.nlm.nih.gov/pmc/articles/PMC6835208/) to be aware how these experiments should be performed or to check if the experiments were performed accordingly.

We apologize for our lack of clarity. Growth curves were performed to compare the baseline growth rates of WT and mutant strains to understand their fitness status. These assays were not performed to evaluate colistin susceptibility/resistance.

We have provided more information for the growth curve methods in lines 108-109, 113, and 152-153:

“Growth curves were performed in mueller-hinton broth with kanamycin (90μg/mL) for plasmid retention.”

Line 113:

Growth curves were performed in mueller-hinton broth.”

And lines 152-153:

Bacteria were grown in mueller-hinton broth with kanamycin (90μg/mL) for plasmid retention when necessary.”